# Risk Factors of Suboptimal Coronary Blood Flow after a Percutaneous Coronary Intervention in Patients with Acute Anterior Wall Myocardial Infarction

**DOI:** 10.3390/jpm13081217

**Published:** 2023-07-31

**Authors:** Natalia Maruszak, Weronika Pilch, Rafał Januszek, Krzysztof Piotr Malinowski, Andrzej Surdacki, Michał Chyrchel

**Affiliations:** 1Faculty of Medicine, Student Scientific Group at Second Department of Cardiology, Jagiellonian University Medical College, ul. Jakubowskiego 2, 30-688 Kraków, Poland; natkamaruszak0@op.pl (N.M.); weronika.pilch@student.uj.edu.pl (W.P.); 2Second Department of Cardiology, Jagiellonian University Medical College, ul. Jakubowskiego 2, 30-688 Kraków, Polandmchyrchel@gmail.com (M.C.); 3Center for Digital Medicine and Robotics, Jagiellonian University Medical College, Kopernika 7E Str., 31-034 Kraków, Poland; 4Department of Bioinformatics and Telemedicine, Jagiellonian University Medical College, Medyczna 7 Str., 30-688 Kraków, Poland

**Keywords:** anterior wall ST segment elevation myocardial infarction, predictors, primary percutaneous coronary intervention, Thrombolysis in Myocardial Infarction flow grade

## Abstract

Background and aims: Primary percutaneous coronary intervention (PCI) is regarded as the most preferred strategy in ST-segment elevation myocardial infarction (STEMI). Thrombolysis in Myocardial Infarction (TIMI) flow grade has been an important and cohesive predictor of outcomes in STEMI patients. We sought to evaluate potential variables associated with the risk of suboptimal TIMI flow after PCI in patients with anterior wall STEMI. Methods: We evaluated 107 patients admitted to our hospital between 1 January 2019 and 31 December 2021 with a diagnosis of anterior wall STEMI treated with primary PCI. Results: Suboptimal TIMI flow grade (≤2) after PCI occurred in 14 (13%) patients while grade 3 was found in 93 (87%) of them presenting with anterior wall STEMI. Failure to achieve optimal TIMI 3 flow grade after PCI was associated with lower TIMI grade prior to PCI (OR 0.5477, 95% CI 0.2589–0.9324, *p* = 0.02), greater troponin concentration before (OR 1.0001, 95% CI 1–1.0001, *p* = 0.0028) and after PCI (OR 1.0001, 95% CI 1–1.0001, *p* = 0.0452) as well as lower mean minimal systolic blood pressure (OR 0.9653, 95% CI 0.9271–0.9985, *p* = 0.04). Conclusions: Among predictors of suboptimal TIMI flow grade after PCI, we noted lower TIMI grade flow pre-PCI, greater serum troponin concentrations in the periprocedural period and lower mean minimal systolic blood pressure.

## 1. Introduction

Percutaneous coronary intervention (PCI) has become accepted as the preferred reperfusion strategy in acute ST-segment elevation myocardial infarction (STEMI) as it results in lower rates of death, reinfarction and stroke compared to fibrinolysis [1,2]. Despite its imperfections, Thrombolysis in Myocardial Infarction (TIMI) coronary flow grade is a validated scale used to assess epicardial perfusion on coronary angiography due to the objectivity of the results in the assessment of blood flow to distal epicardial arterial circulation depending on individual operators [3]. TIMI 0 and 1 flow grade were found as lack of coronary artery patency, TIMI flow grade 2 as impaired blood flow with preserved patency, and TIMI flow grade 3 as normal blood flow [4]. Failure to restore optimal blood flow in the infarct-related artery is not very common and can been observed in 5–23% of patients. Nevertheless, suboptimal coronary blood flow (TIMI ≤ 2) after PCI in patients with acute myocardial infarction is related to higher mortality and worse short- as well as long-term clinical outcomes [5,6,7,8]. The potential mechanisms of non-optimal blood flow post-PCI may be either epicardial coronary artery obstruction or disturbances in microvascular circulation and function of epithelium associated with distal embolisation during PCI, reperfusion injury and inflammatory response [5,9,10]. The anterior wall myocardium is mainly supplied by the left anterior descending coronary artery (LAD). The most common cause of anterior MI is rupture of atherosclerotic plaque in the LAD, which leads to thrombus and a reduction in blood distribution to the heart muscle [11]. In their study, Newman et al. determined that anterior MI represents 32.8% of all STEMIs [12]. The anterior location of MI predisposes to failure in restoring optimal blood flow after PCI and may be associated with worse short- and long-term prognoses compared to inferior or posterior MI [11,13]. It also may be conducive to higher incidences of acute heart failure, ventricular fibrillation and death [11,14].

In the current study, we aimed to assess the potential variables associated with the risk of TIMI ≤ 2 in patients with anterior wall myocardial infarction.

## 2. Materials and Methods

### 2.1. Study Population

This retrospective observational study was performed in the Second Department of Cardiology, University Hospital in Cracow, and included patients presenting with anterior wall STEMI between 1 January 2019 and 31 December 2021 who underwent primary PCI. We excluded patients with MI that were qualified for conservative treatment. Overall, 107 patients were included in this study. Anterior wall STEMI was diagnosed when there were, in at least 2 contiguous leads, ST-segment elevation ≥ 2.5 mm in men < 40 years, ≥2 mm in men ≥ 40 years, or  ≥1.5 mm in women in leads V2–V3 and/or ≥1 mm in the remaining anterior wall leads, according to the guidelines proposed by the European Society of Cardiology [15]. Patients with multivessel coronary disease after index procedure were qualified to the next step of coronary revascularization according to the severity and dissemination of coronary atherosclerosis. The final decision on whether staging was performed during index hospitalization or during subsequent hospitalizations was made during an internal consul, and in selected cases, the consultation was carried out as part of the “Heart-team” in the case of taking into account the patient’s qualification for surgical revascularization, based on the assessment using the SYNTAX score [16]. Patients who had cardiac arrest related to myocardial infarction during hospitalization or in the pre-hospital period were included in this study. This study was approved by the local institutional bioethics committee (Consent number: 1072.6120.336.2022) and institutional review board. Informed written consent was obtained from all patients for the coronary angiography and angioplasty, except for patients who were unable to consent due to their severe state. This study was carried out according to the 1964 Declaration of Helsinki. Informed consent for participation in this study was waived because of the retrospective nature of this study.

### 2.2. Study Protocol

Medical history, physical examination, risk factors, post-procedural ECG, echocardiographic parameters and laboratory test results were obtained retrospectively with the PCI data. Information was gained on the clinical outcomes from hospital discharge summary records and emergency records. Left ventricular function was assessed in B-plane echocardiography according to the Simpson’s rule and current guidelines, and was performed one day after procedure and at the day of discharge [17]. The degree of anterograde arterial flow recovery after the procedure was assessed according to the classification of TIMI and distal embolisation following PCI. Patients were divided into 2 groups, with TIMI flow grade 3 or other than 3 assessed after PCI.

### 2.3. Data Acquisition

STEMI and non-ST-segment elevation myocardial infarction (NSTEMI) diagnoses made according to the International Classification of Diseases 10th Revision (ICD-10) codes were verified by analysis of angiographic images and consulted with operators of individual PCI procedures. Patients with an admission diagnosis of NSTEMI that was verified as anterior wall STEMI during following days of hospitalisation were also enrolled in this study. Category “not classified’’ included patients with negative troponin at admission. While clinically characterising the patients in our study and considering their blood pressure, we considered the minimum and maximum systolic and diastolic blood pressure recorded during the entire hospitalisation period (Table 1). Moreover, analysing biochemical parameters, we included 4 consecutive measurements of troponin levels in each patient, where the first measurement represents the first troponin level determined during hospitalisation. According to the most recent ESC guidelines, we defined heart failure as a clinical syndrome consisting of cardinal symptoms (e.g., breathlessness, ankle swelling, and fatigue) combined with signs (e.g., elevated jugular venous pressure, pulmonary crackles, and peripheral oedema) that result from a structural and/or functional abnormality of the heart associated with elevated intracardiac pressures and/or inadequate cardiac output [18]. In our research, we defined chronic kidney disease as abnormalities of kidney structure or function, present for >3 months with either GFR < 60 mL/min/1.73 m^2^ or markers of kidney damage [19].

### 2.4. Study Endpoints

The principal outcome of interest for the current study was suboptimal coronary blood flow. Suboptimal coronary flow was defined as final TIMI ≤ 2 flow and optimal flow as TIMI 3 flow in the infarct-related artery.

### 2.5. Pharmacological Treatment for Abnormal TIMI Flow Grade after PCI

Strategies adopted for abnormal TIMI flow grade at the time of PCI included intracoronary administration of drugs in regimen, and doses according to operator’s preferences and depending on clinical situation and patient’s condition. Sodium nitroprusside and adenosine were applied in the dose of 200 µg–1 mg. Eptifibatide was administered in a bolus 180 µg/kg bwt and subsequent continuous infusion 2 µg/kg bwt/min. Abcyximab was used in a bolus of 0.25 mg/bwt and afterward in continuous infusion 0.125 µg/kg bwt/min. The dose of nitroglycerin was adjusted to the patient’s clinical state with particular concern for blood pressure values.

### 2.6. Statistical Analysis

Categorical variables are presented as numbers and percentages. Continuous variables are expressed as mean ± standard deviation, and additionally, as median and interquartile range in the case of non-normal data distribution. Normality of distribution was assessed using the Shapiro-Wilk test. Equality of variance was evaluated using Levene’s test. Differences between groups were compared via the Student’s or Welch’s *t*-tests, depending on the equality of variance for normally-distributed variables. The Mann-Whitney U test was applied in the case of continuous variables without normal distribution. Categorical variables were compared via Pearson’s chi-squared or Fisher’s exact test if 20% of cells had a count less than 5. Ordinal variables were compared with the Cochran-Armitage trend test. All baseline/demographic characteristics were used as potential predictors of suboptimal blood flow assessed by TIMI flow grade after PCI in univariable logistic regression models. Multivariable model was not constructed due to the small number of patients with TIMI grade after PCI other than 3. All statistical analyses were carried out using JMP^®^, Version 13.1.0 (SAS Institute Inc., Cary, NC, USA).

## 3. Results

### 3.1. Clinical Characteristics

A total of 107 anterior STEMI patients were enrolled in this study; 14 (13%) had final TIMI flow ≤ 2 (suboptimal flow group) after PCI and 93 (87%) demonstrated a post-PCI TIMI flow grade of 3 (optimal flow group). In the group of patients with suboptimal blood flow after PCI, there were six patients with TIMI flow grade 2, six patients with 2/3, one patient with 0 and one with a 1/2 TIMI flow grade after PCI. The clinical characteristics, laboratory and echocardiographic data for the entire study population are shown in Table 1 and Table 2, respectively. The mean age of patients in the optimal and suboptimal groups was 64.1 ± 14.5 and 65.7 ± 11, respectively. Demographic characteristics were similar between the groups. Sex distribution was also comparable. There were differences between groups in time from onset of chest pain to PCI and treatment with second anti-platelet drugs. Time from onset of pain was longer in TIMI 3 flow grade group. Moreover, percentage of patients treated with second anti-platelet drugs was higher in TIMI grade 3 after the PCI group in comparison to the suboptimal flow group. Furthermore, a subgroup of 14 patients (15.05%) of the TIMI 3 group had previously diagnosed myocardial infarction, whereas no patients of the suboptimal group had MI in the past. Nevertheless, aforementioned differences did not achieve statistical significance. Overall, there were no statistically significant differences in clinical features between the two groups. Table 1. 

### 3.2. Angiographic Data and Primary PCI Procedure

As shown in Table 3, TIMI flow grade before PCI was significantly higher and total stent length was lower in the group that reached optimal TIMI flow grade after PCI, whereas the no-reflow phenomenon occurred only in the suboptimal group. Conversely, there were no statistical differences in the access-site selection, number of stents, balloon/stent ratio, pressure of post-dilatation, duration of the procedure, radiation dose or volume of contrast between the two groups. Table 2. 

### 3.3. Risk Factors of Suboptimal Flow after PCI

Univariable logistic regression analysis allowed the identification of lower minimal systolic blood pressure (OR 0.9653, 95% CI 0.9271–0.9985, *p*-value 0.04), lower TIMI grade before the procedure (OR 0.5477, 95% CI 0.2589–0.9324, *p*-value 0.02), higher troponin 0 level (OR 1.0001, 95% CI 1–1.0001, *p*-value 0.0028) and troponin 2 level (OR 1.0001, 95% CI 1–1.0001, *p*-value 0.0452) as predictors of suboptimal flow after primary PCI in the patient with anterior wall STEMI (Figure 1, Table 4).

## 4. Discussion

The main findings of the current study are that: (1) the incidence of TIMI ≤ 2 was uncommon in patients enrolled in the present study, occurring in 13% of patients with anterior wall myocardial infarction undergoing primary PCI; (2) among predictors of the TIMI flow grade ≤ 2 after PCI was lower minimal systolic blood pressure, greater troponin concentration at baseline and after PCI as well as lower TIMI grade before the procedure and significant predictors of suboptimal flow after primary PCI in the patient with anterior wall STEMI.

Prompt restoration of normal TIMI flow grade 3 in the infarct-related artery is associated with the advantages of reperfusion therapy for ST elevation myocardial infarction [20]. Suboptimal blood flow is considered a multifactorial phenomenon [5]. Due to the fact that thus far, there are limited methods of treatment for poor post-PCI blood flow, it is of significance to assess possible factors that are associated with successful reperfusion. In their study, Elakabawi et al., similarly to us, analysed patients presenting with anterior wall STEMI who had primary PCI, and also suggested that lower systolic blood pressure is a substantial factor predisposing to suboptimal blood flow [5]. Nevertheless, in the aforementioned study, the admission blood pressure was considered, while in our research, we took the minimum systolic blood pressure recorded during the entire hospitalisation period into account [5]. Low systolic blood pressure is a potentially modifiable factor; therefore, proper management of hypotension either with drugs or with mechanical left ventricle support devices such as intra-aortic balloon counter-pulsation (IABP) may be vital for final reperfusion success [5]. Prudent blood pressure reduction among this group of patients in the periprocedural period should also be considered, especially in patients with elevated blood pressure at baseline. This also applies to the administration of any type of vasodilating cocktail to prevent vasoconstriction in patients treated with radial or other peripheral access. In literature on the subject, there are some larger studies, in which their authors evaluated more than 1000 patients, identifying baseline TIMI flow ≤ 1 as an independent predictor of suboptimal TIMI flow grade after PCI in patients with STEMI. This is consistent with our findings [5,8]. The correlation between poor post-PCI blood flow and lower initial TIMI flow may be explained by the higher clot burden or less effective endogenous thrombolysis [8]. Moreover, a high troponin level at admission may be related with tough, more organized, “older” thrombus, which may be challenging for the operator. A low level of troponin, on the other hand, indicates a very early stage of infarction, and thus, a very soft thrombus in most cases. Our data was also consistent with that obtained by Giannitsis et al., who, in their study, concluded that higher TnT is a factor predicting lower rates of post-procedural TIMI 3 flow [21]. Overall, there are reports in which, like in the current study, it has been shown that lower initial TIMI-grade, higher troponin levels and lower minimal systolic blood pressure are important risk factors of developing suboptimal blood flow. These factors may help in the selection of high-risk patients who may require more aggressive adjunctive therapy. In the literature, there is also research in which some other risk factors of suboptimal blood flow have been identified, such as age, diabetes, time from symptom onset to emergency room presentation, number of Q waves on presenting electrocardiogram, left ventricular ejection fraction, thrombus burden grade, total stent length, intravascular ultrasound findings (abnormal lipid pool-like image, lesion elastic), higher Killip class at presentation, and pre-dilatation before stenting as independent predictors of TIMI flow ≤ 2 in patients with myocardial infarction after PCI [5,7,8,22,23]. In the current study, such factors were not demonstrated as statistically significant. On the other hand, in accordance with the study by Guerchicoff et al., earlier reperfusion was not associated with higher rates of final TIMI flow grade 3, which is consistent with the results achieved in our research [1]. Nonetheless, in the previously mentioned study, shorter delay to reperfusion was associated with reduced mortality at 1 year and predisposed to smaller contrast magnetic resonance infarct size [1]. Moreover, in our study, total stent length was lower in the group that reached optimal TIMI flow grade after PCI; however, this difference did not achieve statistical significance. The longer total length of the stents may be conducive to more extensive endothelium damage and promotes microvascular constriction. Furthermore, implantation of stents and balloon dilation may lead to thrombus dislodgement, which results in mechanical obstruction [24]. Glycoprotein IIb/IIIa (GP IIb/IIIa) inhibitors decrease platelet aggregation by preventing fibrinogen from binding to the glycoprotein IIb/IIIa receptors located on the platelet surface [25]. In our study, administration of GP IIb/IIIa was not associated statistically significantly with achieving optimal blood flow after PCI. These may by caused by the fact that, according to guidelines, they are used only in selected patients with STEMI, usually as a bailout therapy for those with complications that occur during percutaneous coronary intervention, rather than as a prophylactic measure [15,26,27]. However, from a practical point of view and our own observations, GPIIB/IIIa inhibitors, apart from adenosine or sodium nitroprusside, are very helpful in regaining blood flow to the vessel periphery in the case of distal embolisation of the coronary microcirculation manifested by no-reflow or slow-flow phenomena. On the other hand, in a meta-analysis aimed at evaluating routine GP IIb/IIIa inhibitor use in STEMI treated with primary percutaneous coronary intervention, their routine use was associated with a significant reduction in rates of TIMI flow < 3 after PCI [26]. Moreover, a subgroup of 14 patients of the TIMI 3 group had previously diagnosed myocardial infarction, whereas no patients of the suboptimal group had MI in the past. Although this difference was not statistically significant, there is the possibility that cardioprotective treatment in patients of the TIMI 3 flow group administered due to history of coronary artery disease could have an impact on outcomes of PCI. In our study, all patients with diagnosed chronic kidney failure did not achieve optimal flow grade after PCI. It is crucial to carefully assess the outcomes of PCI and the strategy of dual anti-platelet therapy (DAPT) in this group [28]. The majority of the aforementioned studies included all patients with STEMI, not only anterior wall MI. The current study was focused on anterior wall MI, as this particular location may predispose to suboptimal blood flow after PCI by itself; thus, research in these population is particularly crucial [13].

### Study Limitations

Our study has several limitations. Firstly, it is a single-centre retrospective study. Therefore, the results may have been affected by some confounding factors and sample size biases. Secondly, comparisons between patients from the optimal and suboptimal groups were not randomised. The number of patients with TIMI flow ≤ 2 was too small to perform multivariable analysis; hence, no independent predictors could be identified. Finally, we did not perform any follow-up examination.

## 5. Conclusions

In the present study, the incidence of suboptimal blood flow in the target artery among patients treated with primary PCI due to anterior wall STEMI assessed as TIMI flow grade ≤ 2 was low and occurred only in 13% of individuals. Suboptimal TIMI flow grade was associated with lower initial TIMI grade, higher troponin levels and lower minimal systolic blood pressure. The results of this small study remain in line with previously published research, and we trust that outcomes of our study will be useful for physicians in treating patients with myocardial infarction undergoing primary PCI and its influence will be mirrored in better treatment results.

## Figures and Tables

**Figure 1 jpm-13-01217-f001:**
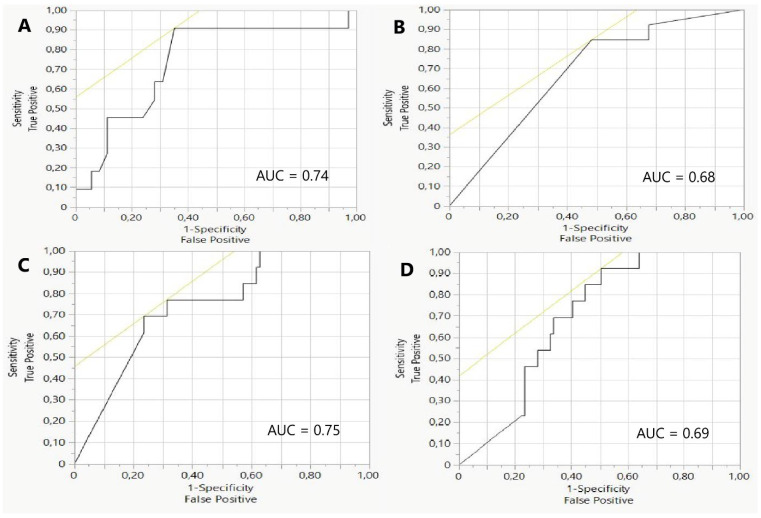
Receiver operating characteristic curves for significant risk factors of suboptimal blood flow assessed by TIMI flow grade (≤2). (**A**) Minimal systolic blood pressure, AUC = 0.74; (**B**) TIMI grade flow before PCI, AUC = 0.68; (**C**) TnI 0, AUC = 0.75; (**D**) TnI 2, AUC = 0.69.

**Table 1 jpm-13-01217-t001:** Clinical characteristics of patients undergoing PCI after anterior wall myocardial infarction.

		Total n = 107	TIMI Grade 3 after PCI n = 93	TIMI Grade Other Than 3 after PCI n = 14	*p*-Value
Age, years		64.32 ± 14.05	64.11 ± 14.49	65.71 ± 11.01	0.69
Gender, male		80 (74.77)	69 (74.19)	11 (78.57)	0.73
BMI, kg/m^2^		26.49 (24.45; 29.47)	26.65 (24.31; 29.41)	26.08 (24.54; 30.2)	0.97
Diagnosis	Anterior wall STEMI	101 (94)	88 (95)	13 (93)	0.22
	Anterior wall NSTEMI	4 (0.04)	3 (0.03)	1 (0.07)	
	Not-classified	2 (0.02)	2 (0.02)	0 (0)	
Hearth failure		50 (46.73)	45 (48.39)	5 (35.71)	0.41
Diabetes mellitus		26 (24.3)	23 (24.73)	3 (21.43)	1
Hypertension		70 (65.42)	62 (66.67)	8 (57.14)	0.55
Kidney failure		10 (9.35)	10 (10.75)	0 (0)	0.35
COPD		6 (5.61)	5 (5.38)	1 (7.14)	0.58
Atrial fibrillation		15 (14.02)	11 (11.83)	4 (28.57)	0.11
Lipid disorders		53 (49.53)	47 (50.54)	6 (42.86)	0.78
Obesity		18 (21.18)	16 (21.92)	2 (16.67)	1
Smoking		31 (28.97)	27 (29.03)	4 (28.57)	1
PAOD		4 (3.74)	4 (4.3)	0 (0)	1
Previous CABG		1 (0.93)	1 (1.08)	0 (0)	1
Previous MI		14 (13.08)	14 (15.05)	0 (0)	0.21
Previous PCI		13 (12.15)	13 (13.98)	0 (0)	0.21
Positive family history		2 (1.87)	2 (2.15)	0 (0)	1
Length of hospitalization, days		8 (6; 10)	7.00 (5; 10)	10.5 (7.5; 15)	0.99
Maximal diastolic pressure, mmHg		80 (70; 90)	80 (70; 90)	80 (70; 100)	0.03
Maximal systolic pressure, mmHg		136.22 ± 25.21	137.04 ± 25.08	130.91 ± 26.64	0.46
Minimal diastolic pressure, mmHg		64.5 (58; 72)	65 (58; 72)	62 (60; 67)	0.91
Minimal systolic pressure, mmHg		107 (94; 119)	108 (96; 120)	96 (90; 100)	0.83
Respirator		10 (9.35)	9 (9.68)	1 (7.14)	1
Second anti-platelet drug		101 (96.19)	89 (97.8)	12 (85.71)	0.09
Statin		103 (98.1)	90 (98.9)	13 (92.86)	0.25
Acetyl-salicylic acid		104 (99,05)	91 (100)	13 (92.86)	0.13
Cardiac arrest		18 (16.82)	16 (17.2)	2 (14.29)	1
Time from onset of pain, min.		175 (110; 332.5)	185 (112.5; 332.5)	97.5 (63.5; 410.5)	0.64
Electrode for temporary pacing		1 (0.93)	1 (1.08)	0 (0)	1

Data are presented as means and SD or median and interquartile range for continuous variables or absolute number and percentages (%) for categorical variables. PCI, percutaneous coronary intervention; BMI, body mass index; COPD, chronic obstructive pulmonary disease; STEMI, ST-elevation myocardial infarction; NSTEMI, non-ST-elevation myocardial infarction; PAOD, peripheral artery occlusive disease; CABG, coronary artery bypass grafting; MI, myocardial infarction.

**Table 2 jpm-13-01217-t002:** Procedural indices.

		Total n = 107	TIMI 3 after PCI n = 93	TIMI Grade after PCI Other Than 3 n = 14	*p*-Value
Coronary arteries angiography		106 (100)	93 (100)	13 (100)	0.07
Radial approach		94 (88.6)	81 (87.1)	13 (100)	0.35
Femoral approach		12 (11.32)	12 (12.9)	0	0.35
Infarct-related artery	LAD	104 (97.2)	90 (96.77)	14 (100)	1
	Cx	8 (7.48)	7 (7.53)	1 (7.14)	1
	RCA	3 (2.8)	2 (2.15)	1 (7.14)	0.35
	SvG Ao	1 (0.93)	1 (1.08)	0 (0)	1
TIMI grade flow before PCI		0	0.5	0	0.0002
		(0; 3)	(0; 3)	(0; 0)	
PCI		107 (100)	93 (100)	13 (92.86)	0.13
CABG		1 (0.93)	0 (0)	1 (7.14)	0.13
Stent		99 (92.52)	86 (92.47)	13 (92.86)	1
Number of stents		1	1	1	0.59
		(1; 1)	(1; 1)	(1; 1.25)	
Number of stents	1 stent	74 (69.16)	64 (68.82)	10 (71.43)	1
	2 stents	23 (21.5)	20 (21.51)	3 (21.43)	1
	3 stents	2 (1.87)	2 (2.15)	0 (0)	1
Total stent length, mm		25	24	28	0.02
		(18; 34.5)	(16; 36)	(23.5; 33)	
Post-dilation, atm		18	19	18	0.4
		(16; 22)	(16.5; 22)	(16; 20)	
Balloon/stent ratio		1.1	1.09	1.17	0.24
		(1; 1.17)	(1; 1.17)	(1.14; 1.2)	
No-reflow		3 (2.83)	0 (0)	3 (23.08)	0.002
Glicoprotein IIb/IIIa inhibitors		53 (49.53)	44 (47.31)	9 (64.29)	0.27
Thrombectomy		39 (36.79)	32 (34.41)	7 (53.85)	0.22
Contrast amount, ml		200	200	200	0.8
		(150; 250)	(150; 250)	(145; 212.5)	
Radiation dose, Gy		0.35	0.38	0.31	0.7
		(0.23; 0.65)	(0.23; 0.66)	(0.15; 0.67)	
Procedure time, min		68.05 ± 31.98	69.58 ± 32.55	57.08 ± 26.03	0.19
ST resolution after primary percutaneous coronary intervention		49 (48.04)	45 (50.56)	4 (30.77)	0.24
Puncture-site-related local complications		4 (3.77)	4 (4.35)	0 (0)	1
Transfusion		6 (5.66)	6 (6.52)	0 (0)	1
Dominant artery	RCA	93 (93.94)	81 (94.19)	12 (92.31)	0.58
	Other arteries	6 (6.06)	5 (5.81)	1 (7.69)	0.79
Qualification for the next stage of revascularisation		24 (22.43)	20 (21.51)	4 (28.57)	0.51

Data are presented as means and SD or median and interquartile range for continuous variables or number and percentages for categorical variables. LAD, left anterior descending; Cx, circumflex artery; RCA, right coronary artery; SvG Ao, saphenous vein grafts to aorta; TIMI, Thrombolysis in Myocardial Infarction; PCI, percutaneous coronary intervention; CABG, coronary artery bypass graft.

**Table 3 jpm-13-01217-t003:** Biochemical and echocardiographic indices.

	Total n = 107	TIMI 3 after PCI n = 93	TIMI Grade after PCI Other Than 3 n = 14	*p*-Value
TnI 0, ng/L	2609	1985.7	25,000	0.95
	(360.84; 25,000)	(315.98; 21,304.52)	(6224.99; 25,000)	
TnI 1, ng/L	9907	8513	23,287.6	0.38
	(2852.26; 25,000)	(2355.31; 23,646.45)	(7908.68; 25,000)	
TnI 2, ng/L	8335.2	6716	15,160	0.2
	(2228.59; 21,388.6)	(1781.09; 18,404.1)	(9248.57; 23,689.23)	
TnI 3, ng/L	3256.5	1946.8	8222.4	0.89
	(909.83; 11,701.49)	(856.14; 10,895.91)	(3567.97; 19,072.32)	
CPK max, U/L	482	432	1467.5	0.21
	(166.5; 1432.5)	(164; 1157)	(685; 2660.5)	
CK-MB max, U/L	109.5	88	222	0.59
	(41.5; 222.5)	(36; 192)	(61; 312)	
Creatinine max, μmol/L	92.2	90.95	109	0.21
	(78.2; 112.5)	(77.65; 110.25)	(84.05; 121.75)	
GFR min, ml/min/1.73 m^2^	75	77	59.50	0.22
	(58; 88)	(59.75; 89)	(53.75; 78.75)	
HB min, g/dL	13.4	13.4	13.05	0.35
	(11.73; 14.78)	(11.95; 15.03)	(11.33; 14.03)	
HT min, %	38.90	38.90	38.35	0.29
	(34.83; 42.38)	(34.98; 42.43)	(32.70; 40.65)	
RBC min [×10^6^]	4.41	4.44	4.11	0.56
	(3.88; 4.81)	(3.90; 4.86)	(3.76; 4.51)	
EF min, %	36.40 ± 10.31	36.81 ± 10.04	33.86 ± 12.01	0.32
EF max, %	38.20 ± 11.11	38.83 ± 10.83	34.21 ± 12.43	0.15
NT-proBNP max, pg/mL	2407	2161	3558	0.41
	(889.75; 11,983.75)	(695.5; 11,776.5)	(1627.5; 14,685.5)	
LVIDd, cm	4.9	4.9	4.8	0.22
	(4.50; 5.25)	(4.58; 5.23)	(3.80; 5.5)	
LVPWd, cm	1	1	1	0.74
	(0.90; 1.05)	(0.9; 1)	(0.95; 1.2)	
TAPSE, cm	2.2	2.2	2.15	0.72
	(2.1; 2.4)	(2.1; 2.4)	(1.78; 2.45)	

Data are presented as means and SD or median and interquartile range for continuous variables or number and percentages for categorical variables. TnI 0, troponin I—1st measurement; TnI 1, troponin I—2nd measurement; TnI 2, troponin I—3rd measurement; TnI 3, troponin I—4th measurement; CPK, creatine phosphokinase; CK-MB, creatine kinase myocardial band; GFR, glomerular filtration rate; HB, haemoglobin; HT, haematocrit; RBC, red blood cells; EF, ejection fraction; NT-proBNP, N-terminal pro-brain natriuretic peptide; LVIDd, left ventricular internal dimension at end-diastole; LVPWd, left ventricular posterior wall thickness at end-diastole; TAPSE, tricuspid annular plane systolic excursion.

**Table 4 jpm-13-01217-t004:** Predictors of non-TIMI 3 grade after PCI in patients with anterior wall myocardial infarction.

	OR	95% CI	*p*-Value
Minimal systolic blood pressure	0.9653	(0.9271; 0.9985)	0.04
TIMI grade before procedure	0.5477	(0.2589; 0.9324)	0.02
TnI 0	1.0001	(1; 1.0001)	0.0028
TnI 2	1.0001	(1; 1.0001)	0.05

TnI 0, troponin I—1st measurement; TnI 2, troponin I—3rd measurement.

## Data Availability

Data could be available on special and reasonable request.

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
