# Peer review of "Risk Factors of Suboptimal Coronary Blood Flow after a Percutaneous Coronary Intervention in Patients with Acute Anterior Wall Myocardial Infarction"

_jpm, 2023, doi:10.3390/jpm13081217_

Round 1

Reviewer 1 Report

Dear authors,

I am glad to review your paper.

The topic is very interesting and the statical analysis is appreciable.

The study is well-written and designed.

Despite it, I have some questions.

First, in the table 1, the percentage of second anti-platelet drugs in group "TIMI grade other than 3 after PCI" is significantly lower than in other group. 

Why didn't you consider this factor in the multivariate analysis or wasn't significative?

Second, how did you define the heart failure?

Third, there were patients in cardiogenic shock? Why didn't you use a left ventricular assist device like Impella to perform the PCI in these patients?

Four, what did you mean with "kidney failure"?  Were there any patients with CKD or was it just AKI? In this concern, I suggest you to add in the discussion the importance to carefully assess the outcomes of PCI and the need for DAPT in patients with CKD, citing Caracciolo A et al "Optimizing the outcomes of percutaneous coronary intervention in patients with Chronic Kidney disease".

Author Response

1.First, in the table 1, the percentage of second anti-platelet drugs in group "TIMI grade other than 3 after PCI" is significantly lower than in other group.

In the table 1 the percentage of second anti-platelet drugs in group TIMI grade other than 3 after PCI is actually lower than in TIMI 3 group , nevertheless this difference didn’t achieve statistical significance (p=0.09).

  1. Why didn't you consider this factor in the multivariate analysis or wasn't significative?

It was not significant in univariate analysis.

  1. Second, how did you define the heart failure

We inserted definition of the heart failure which we used in our research in the data acquisition section.

  1. Third, there were patients in cardiogenic shock? Why didn't you use a left ventricular assist device like Impella to perform the PCI in these patients?

This is how some patients were after cardiac arrest, some of them in a serious condition, however, in this group of patients we did not use Impella, in the case of acute conditions we use it in exceptional situations, much more often we do it in elective cases in patients with reduced LVEF.

5.,, Four, what did you mean with "kidney failure"? Were there any patients with CKD or was it just AKI? In this concern, I suggest you to add in the discussion the importance to carefully assess the outcomes of PCI and the need for DAPT in patients with CKD, citing Caracciolo A et al "Optimizing the outcomes of percutaneous coronary intervention in patients with Chronic Kidney disease"

We inserted appopirate definition in the data acquisition section.As ,,kidney failure’’ we considered only patients with a chronic kidney disease, not with AKI.

Reviewer 2.

The definition of kidney failure (better renal failure) should be given

We inserted appropriate definition in the data acquisition section.

 It is strange that the maximal diastolic pressure has exactly the same mean values, but p-value is 0.03.

This is not maximal, but Q1 and Q3. Mean is 80.72 (±12.90) vs. 86.00 (±23.55) while min. and max. is 56.00 / 116.00 and 61.00/145.00.

 The fact that patients with final TIMI<3 had longer stents should be discussed.

Thank you very much for this advise. We referred to total stent length in discusssion section.

It is not clear for me also, what factors were included into multivariate model for prediction of final TMI III.

This is explainde into the Statystical Analysis section. If necessary we could provide univariate analysis- table.

The language in general is good, the definite statements should be escaped for such a small sample size.

In study limitation section there is a relevant comment about sample size bias.

Reviewer 2 Report

The manuscript in general is ok. The definition of kidney failure (better renal failure) should be given. It is strange that the maximal diastolic pressure has exactly the same mean values, but p-value is 0.03. The fact that patients with final TIMI<3 had longer stents should be discussed. It is not clear for me also, what factors were included into multivariate model for prediction of final TMI III.

The language in general is good, the definite statements should be escaped for such a small sample size.

Author Response

(The authors gave the same response as above.)

Round 2

Reviewer 1 Report

Dear Authors

I appreciated your efforts to reply to my questions